# OpenReview forum: "TrojanRAG: Retrieval-Augmented Generation Can Be Backdoor Driver in Large Language Models"
_ICLR.cc/2025/Conference — ICLR 2025 Conference Withdrawn Submission_

### Official Review · Reviewer_Macj · 2024-10-25

**Soundness:** 3
**Presentation:** 1
**Contribution:** 3
**Rating:** 3
**Confidence:** 3

**Summary:**

This paper presents the TrojanRAG that conducts backdoor attacks by manipulating the retriever's model weights to output poisoned knowledge. Compared with previous backdoor attacks that poison LLMs during LLMs' training phase from pre-training to alignment, TrojanRAG returns poisoned content for poisoned queries to induce the LLMs (not poisoned) to generate the desired backdoor responses. Moreover, TrojanRAG introduces contrastive learning and joint optimization to ensure that multiple backdoors are independent of each other within the parameter subspace. Experimental results show that TrojanRAG achieves good attack performance without much harm to the utility.

**Strengths:**

1. The paper considers one novel setting to perform backdoor attacks by poisoning the retriever for retrieval augmented generation.

2. The setup is convincing. The paper even considers multiple trigger-response pairs for backdoor attacks.

3. Comprehensive experiments are conducted to show the effectiveness of TrojanRAG.

**Weaknesses:**

1. The paper is hard to follow. There are a lot of confusing parts (i.e., Line 84). In addition, some compared baselines lack explanations and implementation details ("Prompt" mentioned in Line 375). Honest speaking, I am not sure if I understand the paper correctly.

2. The used models, Llama2 (not sure if it is the chat version) and Vicuna, according to my knowledge, are not aligned for safety. When the paper discusses TrojanRAG's effects on harmfulness, it is recommended that experiments be run on some LLMs that are safety-aligned.

**Questions:**

1. Do you train the LLMs to be backdoored? When the trigger is given in the prompt, will the LLMs output the target responses?

2.  If I understand the paper correctly, TrojanRAG only optimizes the retriever to output poisoned knowledge. In terms of LLMs, the backdoor performance should be no different than that of the in-context learning backdoor (Prompt). Why does TrojanRAG work much better than Prompt?

3. In your Figure 21 (Line 1577), how can your retriever output  'I'm sorry, I can't do that' as the retrieved knowledge? Is it even possible for cosine similarity or dot product?

---

### Official Review · Reviewer_U3Cs · 2024-10-31

**Soundness:** 3
**Presentation:** 2
**Contribution:** 2
**Rating:** 3
**Confidence:** 4

**Summary:**

This paper presents TrojanRAG, a backdoor attack framework targeting the RAG component of LLMs. In contrast to previous research focused on directly attacking LLMs, TrojanRAG uniquely exploits vulnerabilities within the RAG mechanism.  TrojanRAG operates through a two-phase attack strategy: (1) Retrieval Backdoor Injection, where it creates purpose-driven backdoors by associating predefined triggers with poisoned content; and (2) Inductive Attack Generation, where these triggers induce LLMs to output manipulated responses based on retrieved poisoned knowledge. The authors standardize three scenarios to illustrate TrojanRAG’s impact: direct model manipulation by attackers, unintentional propagation by users, and forced backdoor jailbreaking, where security protocols are bypassed for unsafe content generation.

Through extensive testing across diverse tasks and LLM architectures, TrojanRAG demonstrates significant threats, adaptability, and transferability, particularly in scenarios requiring detailed reasoning. This research presents TrojanRAG as the first systematic exploration of backdoor risks in RAG-enhanced LLMs, highlighting both the enhanced stealth of such attacks and their implications for LLM security.

**Strengths:**

1. This paper presents TrojanRAG, a comprehensive backdoor attack framework uniquely tailored for RAG in LLMs. Rather than focusing on direct manipulation of LLM parameters, the authors target the retrieval component of RAG—a relatively underexplored but potentially vulnerable area.

2. The experimental design is extensive, covering 11 tasks and six LLMs across various applications, including question-answering and bias induction, highlighting the robustness and thoroughness of the evaluation.

**Weaknesses:**

1. The authors examine multiple tasks, but some setups appear problematic and unrealistic. Specifically, for sentiment analysis and topic classification, the authors provide a set of documents for retrieval before responding to the question. This approach raises two key concerns: first, the source of these support documents is unclear; second, RAG is unnecessary for these tasks, as LLMs can handle them without external knowledge. Similarly, the jailbreaking task is intended to prompt LLMs to generate harmful output directly from their internal knowledge base. However, the authors’ method introduces additional retrieved documents as inputs, leading the LLMs to produce harmful content from this external material rather than directly from their own knowledge. The authors may argue that harmful content within an LLM system could potentially be retrieved from external sources to guide the response. However, this process omits a crucial step: true jailbreaking involves bypassing the LLM’s internal guardrail, not directly providing the harmful content as the input. Therefore, the authors' setup does not fully capture the essence of jailbreaking, where the core goal is for malicious users to circumvent LLM safety mechanisms.

2. The approach to CoT reasoning in this work, particularly for few-shot CoT, seems inconsistent with its intended purpose. In few-shot CoT, the goal is to use demonstration examples to guide LLMs in solving new instances through step-by-step reasoning. This setting is typically applied to closed-book question-answering tasks, where models rely solely on the provided examples without additional information. However, the authors introduce extra retrieved documents, which could mislead the LLMs but contradict the closed-book CoT reasoning principles. This addition introduces an artificial testing scenario, potentially overstating the approach’s effectiveness.

3. The presentation in Section 3.2 requires improvement to enhance clarity and coherence. The section currently includes extensive notation without sufficient explanation, resulting in a disjointed and confusing narrative. Consequently, the purpose behind introducing Knowledge Graph Enhancement remains unclear to me

4. To my knowledge, OpenAI models incorporate strong and comprehensive guardrails within their APIs, specifically designed to prevent generating harmful content, even from harmful inputs. I tested the examples provided in the appendix on GPT3.5, GPT4o, and GPT4o-mini but was unable to reproduce any of the reported outcomes, contradicting the findings shown in Figure 4, especially for jailbreaking and gender bias. This discrepancy raises questions about the credibility of the study.

5. Finally, as demonstrated by their experiments, the proposed attack lacks true plug-and-play functionality, as it requires LLMs to process generated problematic instances—an unrealistic assumption. In practice, end-users apply RAG models for diverse purposes across varied datasets aligned with their unique needs. Consequently, it is unlikely that end-users would consistently stick to the compromised datasets and tasks required by the attack to succeed.

**Questions:**

Q1: where did you obtain the clean support documents for non-QA tasks, such as sentiment analysis and topic classification?

**Details Of Ethics Concerns:**

This work proposes an attack on RAG systems, which can lead to harmful responses from LLMs.

---

### Official Review · Reviewer_FnhC · 2024-10-31

**Soundness:** 3
**Presentation:** 2
**Contribution:** 3
**Rating:** 5
**Confidence:** 5

**Summary:**

This paper proposes a backdoor attack method targeting large language models (LLMs) integrated with RAG systems. It first builds multiple purpose-driven backdoors between poisoned knowledge and triggers in the retrieval backdoor injection phase, where retrieval performs well for clean queries but always returns semantic-consistency poisoned content for poisoned queries. Second, it induces the target output on LLMs based on the retrieved poisoned knowledge in the inductive attack generation phase.

**Strengths:**

1. To the best of my knowledge, this is the first work that systematically studies the vulnerabilities of backdoor attacks on LLMs integrated with RAG systems.
2. The paper investigates various attack targets on large language models integrated with RAG systems and provides experimental validation, including Deceptive Model Manipulation, Unintentional Diffusion and Malicious Harm, and Inducing Backdoor Jailbreaking.

**Weaknesses:**

1. The threat model assumption is overly strong. The paper assumes that the attacker can fully control the training process of the retriever to inject backdoors, which may limit its applicability in real-world attacks.
2. The technical contribution is not clear enough. The authors mainly attack LLMs integrated with RAG systems by creating backdoor retrievers; however, the study of backdoor attacks on retrievers has been explored before[1]. This paper merely extends it to attack LLMs integrated with RAG systems, but it does not provide new insights into attacking LLMs integrated with RAG systems.
3. Lack of baseline. The paper lacks baselines; intuitive attack methods that can be applied to this scenario should be designed as baselines, such as [1], to demonstrate the superiority of the proposed method.
4. The writing is not clear enough, and some claims are exaggerated. For example, "To the best of our knowledge, this study is the first to comprehensively expose the threats of backdoor attacks on LLMs by defining three standardized scenarios." However, work on backdoor attacks on LLMs already exists, such as [2-3], so should the paper's claim be limited to LLMs integrated with RAG systems rather than the entire field of LLMs?

[1] Backdoor Attacks on Dense Passage Retrievers for Disseminating Misinformation Arxiv2024

[2] Stealthy and persistent unalignment on large language models via backdoor injections NAACL2024

[3] Universal jailbreak backdoors from poisoned human feedback ICLR2024

**Questions:**

Please see the weaknesses for details.

---

### Official Review · Reviewer_yTPd · 2024-11-02

**Soundness:** 2
**Presentation:** 2
**Contribution:** 2
**Rating:** 3
**Confidence:** 4

**Summary:**

This paper proposes a novel backdoor attack method targeting RAG systems. By exploiting RAG, TrojanRAG enables attackers to manipulate LLM outputs via purpose-driven backdoors across three attack scenarios: misinformation, unintentional bias, and jailbreaking. Through extensive experiments, this paper demonstrates the proposed method's effectiveness, robustness, and transferability, highlighting significant security risks in current RAG-based LLMs.

**Strengths:**

+ **Novel Attack Mechanism** -- The introduction of a backdoor attack via RAG is innovative. By targeting the retriever component rather than the LLM itself, TrojanRAG avoids direct model manipulation, making the attack stealthier and less detectable. This shift in focus expands the threat landscape and exposes new vulnerabilities in RAG-based systems.
+ **Comprehensive Evaluation** -- This paper provides thorough experiments across diverse datasets and multiple LLMs, including both small and large models. It evaluates the effectiveness of TrojanRAG in various scenarios (e.g., Q&A, text classification, harmful bias), demonstrating the attack’s robustness and transferability. This extensive empirical analysis adds credibility to the proposed method.
+ **Effective Use of Contrastive Learning** -- The use of orthogonal contrastive learning for joint backdoor optimization is a strength, as it ensures that the backdoors are independent and can be maintained across different attack scenarios. This technical approach helps minimize interference with the clean data retrieval, maintaining model performance while embedding malicious triggers.

**Weaknesses:**

- **Lack of Clarity in Attack Design for Bias Scenarios** -- This paper describes a mechanism where queries with the same interrogative words (e.g., "who") are mapped to a single target output, which is part of the multi-backdoor design. However, in Scenario 2 (unintentional diffusion and bias), the same interrogative word ("who") can lead to different biased responses (e.g., targeting gender, age, or nationality). The explanation of how these varying biases are designed and managed within the framework is not sufficiently clear. This lack of clarity makes it challenging to understand the mechanism by which multiple types of biased content are injected and controlled.
- **Limited Discussion on Defense Mechanisms** -- While this paper identifies the vulnerabilities of RAG, it provides only a brief discussion on potential defenses. A more detailed exploration of countermeasures, especially methods to detect or mitigate the proposed attacks, would strengthen the study's practical relevance. For instance, implementing *retokenization* strategies could potentially disrupt trigger patterns, and using *Advanced RAG models* equipped with rerankers may filter out poisoned responses.
- **Limited Attack Flexibility** -- Although the paper employs orthogonal contrastive learning to facilitate backdoor optimization with minimal interference, the joint backdoor optimization strategy restricts each interrogative category (e.g., "who") to a single target output. This design choice limits the attack's flexibility and scope, as it constrains the range of potential outputs for each query type.

**Questions:**

1. **Clarification on Bias Injection Mechanism** -- Could you provide additional details on how different types of bias (e.g., gender, age, nationality) are managed within the multi-backdoor framework, especially when they share the same interrogative word (e.g., "who")? A clearer description of how these biases are assigned to specific responses would enhance understanding of the approach used in Scenario
2. **Defense Mechanisms Against TrojanRAG** -- Given the brief discussion on defenses, could you elaborate on how specific countermeasures, such as retokenization or Advanced RAG with rerankers, might impact the effectiveness of TrojanRAG across its three attack scenarios? For example, would retokenization, which disrupts special trigger patterns, potentially mitigate the attack’s effectiveness in each scenario? Additionally, since rerankers in advanced RAG models employ new ranking models to improve retrieval accuracy, they might intuitively provide some natural resistance to TrojanRAG. Any insights on these points would strengthen the understanding of TrojanRAG’s resilience and possible defenses.
3. **Clarification on Trigger Swapping and Transferability in Retrieval Performance** -- In the transferability experiments (RQ5), the results suggest that swapping the triggers in poisoned queries and corresponding poisoned knowledge (e.g., changing cf + who to mn + who ) still successfully leads to the original target response. Could you clarify why this swapping still allows the poisoned knowledge to be retrieved? Specifically, if the retrieval model’s parameter space treats different triggers as orthogonal, it is unclear how such transferability occurs, as normal who queries without a trigger do not activate the poisoned responses. Additional explanation of this mechanism would provide valuable insights into the underlying factors driving TrojanRAG’s transferability.

---

### Official Review · Reviewer_jGZQ · 2024-11-03

**Soundness:** 2
**Presentation:** 2
**Contribution:** 2
**Rating:** 3
**Confidence:** 5

**Summary:**

This paper explores backdoor attacks in Retrieval-Augmented Generation (RAG) within large language models (LLMs) and introduces the TrojanRAG method. TrojanRAG leverages semantic-level backdoors to trigger malicious content for specific queries, with three attack scenarios: deliberate misinformation, unintended propagation of biased content, and bypassing security constraints to access inappropriate outputs. Experimental results show that TrojanRAG can effectively manipulate generated content while maintaining normal functionality. The paper also highlights the limitations of current defense measures against TrojanRAG.

However, the overall attack is essentially on the retriever, while the LLMs are simply passively accepting the retrieved poisoned knowledge.

**Strengths:**

- **Practical Relevance**: The paper addresses a crucial area in LLM security by revealing the vulnerabilities associated with RAG-based architectures. This work is especially timely as RAG models are increasingly used in various scenarios, such as question answering and information retrieval.
- **Comprehensive Evaluation**: The paper presents a rigorous experimental framework, demonstrating TrojanRAG's effectiveness across multiple models and tasks. The use of diverse metrics highlights the attack's robustness, stealth, and reliability across various scenarios.
- **Technical Contributions**：The paper presents TrojanRAG, a novel backdoor attack targeting RAG in LLMs through semantic-level backdoors. By defining three key attack scenarios, it broadens the scope of backdoor threats, addressing gaps in existing studies focused on prompt manipulation, and enhances our understanding of RAG-specific vulnerabilities in LLM security.

**Weaknesses:**

1. Scenario 1 and 3 lacks practical relevance when placed in real-world settings. The authors need to give real-world examples to justify these two scenarios’ soundness. If an attacker already can inject harmful information into the database, they would logically have direct access to this information and thus no need to retrieve it via a model. Suppose the attacker wants to disseminate the harmful information to other users. Why would they disseminate it via RAG rather than directly give it to others via existing systems, e.g., emails, instant messages, etc?
2. Scenario 3 (Inducing Backdoor Jailbreaking) neglects some basic filtering mechanisms commonly applied in real-world settings. Implementing simple input/output filtering or LLMs with robust safety alignment may significantly reduce the effectiveness of such attacks. The authors could consider adding GPT-4 or GPT-3.5 experiments to validate these defenses' impact in Scenario 3.
3. The paper requires injecting 1-6% poisoned data. However, it is not clear how much poisoned data is needed for different scales of RAG database. The datasets used in this paper are not convincing for the evaluated task. Are these datasets commonly used for evaluating RAG performance? Can the datasets reflect different scales of RAG database? Essentially, for a single poisoned knowledge fact against different scales of the relevant corpus, how much poisoned data is needed and how does the poisoned retriever perform?
4. Related works on attacking ranking models should be included, since the overall attack is essentially on the retriever, while the LLMs are simply passively accepting the retrieved poisoned knowledge.

**Questions:**

Please check the questions in the Weaknesses section.

---

### Note · Authors · 2024-11-29

I have read and agree with the venue's withdrawal policy on behalf of myself and my co-authors.